# Integrated Proteotranscriptomics Reveals Differences in Molecular Immunity between Min and Large White Pig Breeds

**DOI:** 10.3390/biology11121708

**Published:** 2022-11-25

**Authors:** Liyu Yang, Xin Liu, Xiaoyu Huang, Na Li, Longchao Zhang, Hua Yan, Xinhua Hou, Lixian Wang, Ligang Wang

**Affiliations:** 1Key Laboratory of Farm Animal Genetic Resources and Germplasm Innovation of Ministry of Agriculture of China, Institute of Animal Science, Chinese Academy of Agricultural Sciences, Beijing 100193, China; 2College of Animal Sciences, Shanxi Agricultural University, Taigu 030800, China; 3Jilin Academy of Agricultural Sciences, Changchun 130033, China

**Keywords:** transcriptome, proteome, cytokines, min pigs, disease resistance

## Abstract

**Simple Summary:**

The immune system plays a vital role in immune defense, stability, surveillance, tolerance, and regulation. Long-term selection or evolution is an important factor governing the development of disease resistance in pigs. To better understand the molecular mechanisms underlying different levels of disease resistance, we used transcriptomics and proteomics analysis to analyze the immune differences between Chinese and foreign pig breeds and thereby identify critical genes and proteins involved in immune regulation. Key genes involved in these pathways could act as candidates for the discovery of useful polymorphisms that could be included in breeding programs to enable the selection of naturally resistant animals. The results of this study will provide new insights into breeding pigs for disease resistance.

**Abstract:**

Long-term selection or evolution is an important factor governing the development of disease resistance in pigs. To better clarify the molecular mechanisms underlying different levels of disease resistance, we used transcriptomics and proteomics analysis to characterize differences in the immunities between six resistant (Min pig) and six susceptible (Large White, LW) pigs which were raised in the same environment. A total of 135 proteins and 791 genes were identified as being differentially expressed between the Large White and Min pig groups. Protein expression clustering and functional analysis revealed that proteins related to immune system process, humoral immune response, the B cell receptor signaling pathway, lymphocyte-mediated immunity, and innate immune responses were more highly expressed in Min pigs. Transcriptome gene set enrichment analysis was used to reveal that pathways of cell adhesion molecules and antigen processing and presentation are significantly enriched in Min pigs. Integrated proteomics and transcriptomics data analysis identified 16 genes that are differentially expressed at both the mRNA and protein levels. In addition, 13 out of these 16 genes were related to the quantitative trait loci of immune diseases, including neural EGFL-like 2 (*NELL2*) and lactate dehydrogenase B (*LDHB*), which are involved in innate immunity. Correlation analysis between the genes/proteins and cytokines shows upregulated proteins in LW pigs in association with immunosuppressive/pro-inflammatory cytokines, such as interleukin (IL) 10, IL6, and tumor necrosis factor alpha. This was further validated using parallel reaction monitoring analysis. In summary, we discovered several potential candidate pathways and key genes/proteins involved in determining differences in disease resistance between the two studied pig breeds, which could provide new insights into the breeding of pigs for disease resistance.

## 1. Introduction

Pig farm diseases, especially viral infectious diseases, seriously threaten the health of pigs and benefits of pig production. Immunity is not only a physiological reaction of animals but also a defense mechanism formed by evolution. The immune system is divided into innate and acquired immunity and plays a vital role in immune defense, stability, surveillance, tolerance, and regulation [1]. When pathogenic microorganisms invade organisms, the body can resist them by activating the immune system [1]. The immune system uses the coordinated activities of innate and adaptive immune system components to recognize and eliminate pathogens, reduce the host’s pathogen burden, alleviate the dysfunction of parenchymal tissues, and enhance the host’s disease resistance [2]. The results of previous studies have demonstrated considerable differences in disease resistance between Chinese and foreign pig breeds [3,4]. In 2006, a highly pathogenic form of north American PRRSV (HP-PRRSV) emerged in China. The survey data showed that the Tongcheng, Laiwu, and Lulai Black pigs have low mortality rates of 0, 0, and 0.6%, respectively. However, the mortality rate of the Large White (LW) is as high as 54%, suggesting there may be differences in immunity between Chinese and foreign pig breeds [5]. The Min pig (M) is an excellent local breed in China, with the characteristics of high immunity and substantial disease resistance advantages, and is affinis to Laiwu pigs [6]. Enhancing resistance and tolerance to pathogens remains an important objective during selection in the production of livestock animals. Therefore, it is necessary to understand the molecular and genetic mechanisms of the immune system in different pig breeds, which is of great value for breeding pigs with high disease resistance.

In previous studies, the porcine immune system has been investigated by transcriptome and proteome profiling for researching disease resistance in pigs [7,8,9,10,11]. Mehrotra et al. generated the blood mononuclear cell transcriptomes of the different pig breeds Ghurrah and Landrace to elucidate the mechanisms of host immune response against CSF vaccination [10,12]. Ni et al. researched the transcriptome profiling of porcine lung from two pig breeds in response to *Mycoplasma hyopneumoniae* infection [13]. In the research of Ling et al., the authors examined transcriptome and proteome differences of lymph nodes between Tongcheng and Large White pigs [14]. We also have observed an interesting phenomenon that the Min pigs have lower mortality than LW pigs in our farm. Additionally, when the disease struck the farm, the growth or reproductive performance of the large white pigs were severely affected, while the Min pigs in the same enclosure remained normal. However, there are few reports describing integrated analysis of protein and mRNA associated with the blood and serum of Chinese domestic and foreign pigs under common raising conditions.

In this study, we simulated the natural state of the breeding environment in China to evaluate the immunity differences between M and LW pigs. We used proteomics and transcriptome analysis to identify global mRNA and protein expression patterns in whole blood and serum collected from different pig breeds for 80 days. Critical genes and proteins involved in immune regulation were also identified. Key genes involved in these pathways could act as candidates for discovering polymorphisms, which could be included in breeding programs to enable the selection of naturally resistant animals. Our results resent a systems-level comparison of indigenous and commercial breed groups regarding the effects on the proteome and transcriptome when subject to the same environmental conditions. The results are helpful in further understanding the molecular mechanism of disease resistance in pigs and provide a valuable reference for breeding of pigs for disease resistance.

## 2. Materials and Methods

### 2.1. Subjects and Tissue Collection

We selected 17 piglets from 3 littermates of different pig breeds (Min pig (*n* = 6), Large White (*n* = 6), and cross-bred of LW×M (C, *n* = 5)) from Changping Breeding Pig Farm, Institute of Animal Sciences, Chinese Academy of Agricultural Sciences (Beijing, China). From each litter, piglets were selected based on their weight at 28 days, avoiding underweight or overweight individuals. All of the piglets were raised for 35 days in the farrowing house and 45 days in the nursery house. These pigs were free to eat and drink until the end of the 80 days. Additionally, the average weight of 80-day-old M, LW and C pigs were 25.2 ± 2.3 kg, 33.4 ± 2.9 kg and 28.3 ± 3.1 kg, respectively. Whole blood and serum samples were collected and stored at −80 °C until use.

### 2.2. Sample Preparation

Serum (10 μL) was added to lysate (0.1 M Tris-HCl, 8 M urea, 1% SDS, pH 8.0) of 40× vol. The protein concentration of the resulting sample solution was determined using a Bicinchoninic Acid (BCA) FP0010 protein kit (Sigma-Aldrich, Taufkirchen, Germany). DTT was added to the sample for reaction at 37 °C for 1 h, followed by 60 mM IAA with reaction for 30 min. After the reaction, 6× vol of cold acetone was added and allowed to precipitate at −20 °C for 5 h. The precipitated protein was collected and cleaned using cold acetone. Next, 20 ng/μL trypsin 50 μL was added, and the sample was digested at 37 °C overnight. Finally, the enzyme-cut polypeptide was desalted using a C18 column before further analysis.

### 2.3. Construction of the Data-Independent Acquisition (DIA) Spectral Library and Library Searches

The mixture of peptides was redissolved in Solvent A (Solvent A: 20 mM ammonium formate aqueous solution, ammonia adjusted to pH 10.0). A reverse column (Accucore C18 column, 2.1 × 150 mm, Ultimate 3000 system, Thermo Fisher Scientific, Waltham, MA, USA) was connected for high pH separation. The column was balanced for 15 min under the initial conditions, the flow rate was maintained at 0.3 mL/min, and the column temperature was maintained at 30 °C. A total of 10 fractions were collected. The desalted, lyophilized peptide was redissolved in solvent C (C: 0.1% formic acid aqueous solution) and analyzed by LC–MS/MS. The mass spectrum parameters were set as follows: (1) MS: scan range (m/z) = 350–1500; resolution = 60,000; AGC target = 4 × 10^5^; maximum injection time = 50 ms; included charge states = 2–6; (2) HCD–MS/MS: resolution = 30,000; isolation window = 4 Da; AGC target = 5 × 10^4^; maximum injection time = 120 ms; collision energy = 35. The original data were analyzed and searched in Spectronaut 13 (Biognosys AG, Schlieren, Switzerland), and the database was built using the default software parameters. The false-positive rate (FDR) of both parent ions and peptides was set at 1%.

### 2.4. DIA Data Acquisition and Analysis

A 30 μL aliquot of solvent A (A: 0.1% formic acid aqueous solution) was added to each sample to prepare suspension A. Solvent A (9 μL) was added to 1 μL 10× iRT peptide. The mixture was separated with Nano-LC and analyzed online using electrospray ionization tandem mass spectrometry (Thermo Fisher Scientific, Waltham, MA, USA). Sample aliquots of 3 μL were taken (analysis column: Omics High-Resolution Series Monolithic capillary HPLC columns, 100 μm × 50 cm, KYOTO MONOTCHE) using a gradient of 120 min for sample separation. The column flow was controlled at 600 nL/min, the column temperature was 45 °C, and the electrospray voltage was 2.2 kV. DIA data were analyzed using Spectronaut (BGS Factory Settings (Default)). IRT peptide software was used to correct the retention time and quality window (protein qualitative criteria: precursor threshold 1.0% FDR, protein threshold 5.0% FDR). The mutation strategy generates a decoy database. Spectronaut automatically corrected all MS1 data that met the screening criteria and were used to calculate expression levels, excluding all interference debris ions. The average peak areas of the first three peptides with less than 1.0% FDR were used for protein group quantification. The data presented in the study were deposited in the OMIX repository, accession number OMIX001648.

### 2.5. Transcriptomic Analysis

Twelve whole blood samples were collected from 80-day-old Large White and Min pigs. A NanoDrop 2000 spectrophotometer (Thermo Scientific, Wilmington, DE, USA) and an Agilent 2100 Bioanalyzer (Agilent Technologies, Santa Clara, CA, USA) were used to detect total RNA sample purity, concentration, and integrity. Total RNA was isolated using the improved cetyltrimethylammonium ammonium bromide (CTAB) method. Whole white blood cells (WBC) were used for RNA extraction. Takara RNAse-free DNAse I (Dalian, China) was used to process the total RNA from each replicate to remove genomic DNA. Cleaved RNA fragments were reverse transcribed to create the cDNA library. The Illumina PE150 platform (San Diego, CA, USA) was employed to generate paired-end reads and analyze the transcriptome of 12 blood samples. The data presented in the study were deposited in the GSA repository, accession number CRA007900.

### 2.6. Differentially Expressed Protein (DEPs) and Gene Bioinformatics Analysis

Differentially expressed (DE) proteins in C vs. M, C vs. LW, and LW vs. M were identified using two-sided Student’s *t*-test based on two criteria of fold change (FC) > 1.2 or FC < 0.83 in expression levels and *p*-value ≤ 0.05. Gene ontology (GO) terms in biological process (BP), cellular component (CC), and molecular function (MF) categories were enriched using KOBAS 3.0 (Beijing, China), using *p*-value < 0.05 as a threshold. Kyoto encyclopedia of genes and genomes (KEGG) analysis was conducted using KOBAS 3.0, and *p*-value < 0.05 was considered to indicate significance.

### 2.7. Gene Set Enrichment (GSEA) Analysis

GSEA is an aggregate score and running-sum statistic approach that enables molecular-signature-based statistical significance testing that considers the entire gene set and contains a ranked list of all expression values in a data set without requiring a cutoff of differentially expressed values for functional analysis. We performed gene set enrichment analysis (GSEA) using the R package. We used the Molecular Signatures Database (MsigDB, http://software.broadinstitute.org/gsea/msigdb, accessed on 12 September 2022) C2. KEGG collection gene sets to evaluate relevant pathways and molecular mechanisms. We set the minimum and maximum gene sets as 5 and 5000, respectively, based on gene expression profile grouping. We considered | NES | > 1, *p*-value < 0.05, FDR < 0.25 as significant gene sets for GSEA enrichment analysis. The longest red line in the figure was the peak line, which was divided into two sides. We called the right side the leading edge and named the genes on this side as leading targets, which were core genes worth exploring.

### 2.8. Protein–mRNA Correlation Analysis

The nine-quadrant tool (https://www.omicshare.com/tools, accessed on 12 September 2022) was used to analyze the correlations of protein–mRNA pairs for all of the expressed proteins and mRNAs. The regularized-logarithm transformation (rlog) values of expression levels and global Pearson correlation coefficients were calculated for 16 protein–mRNA pairs within Min and LW serum. mRNA and protein expression levels were generated using the R package *p*-value ≥ 0.05: no marks; *p*-value < 0.05: *; *p*-value < 0.01: **; *p*-value < 0.001: ***).

### 2.9. Differentially Expressed Genes (DEGs) and Quantitative Trait Locus (QTL) Co-Location Analysis

The Animal QTLdb is open and offers dynamic, updated, publicly available trait-mapping data to locate and compare discoveries within and between species. A total of 35,384 QTLs from 762 publications containing 716 phenotypic traits have been collected in the current release of the Pig QTLdb (https://www.animalgenome.org/cgi-bin/QTLdb/SS/index, accessed on 12 September 2022). We compared the DEGs with QTLs in the pig QTLdb and previous reports of the immune trait to screen the DEGs for the candidate genes associated with pig immunity.

### 2.10. Parallel Reaction Monitoring (PRM) Analysis

The peptide digests were separated and analyzed using Easy nLC 1200 (Thermo Scientific, Waltham, MA, USA) and Q-Exactive HFX (Thermo Scientific, Waltham, MA, USA), respectively. The target protein data were imported into the Skyline 4.1 software (Sciex, Toronto, ON, Canada). The raw file was imported into the original data of data dependent acquisition (DDA) by MaxQuant software (Martinsried, Germany) to screen the matching peptides with high ionic strength and fewer heteropeaks. Database retrieval parameters were set as follows: missed cleavage was set to 0, and the reliability of the peptide was ≥0.95. Finally, the database search results were imported into Skyline software to compare the selected candidate peptides.

### 2.11. Quantification of Cytokines in Serum

A porcine cytokine array was used to quantify the different responses between M and LW pigs to immunization. A total of 50 μL serum of each sample was taken for the multiplex assay with a porcine (pig) Cytokine Array 1 Kit (Raybiotech, Peachtree Corners, GA, USA) according to the manufacturer’s instructions [15].

### 2.12. Correlation Analysis between the Genes/Proteins and Cytokines

OmicStudio tools (https://www.omicstudio.cn/tool, accessed on 12 September 2022) were used to calculate the correlation of continuous data between 16 differentially expressed proteins and cytokines based on Pearson correlation (*p*-value ≥ 0.05: no marks; *p*-value < 0.05: *; *p*-value < 0.01: **; *p*-value < 0.001: ***).

### 2.13. Statistical Analysis

Statistical analysis was performed with the SPSS 20.0 software (Chicago, IL, USA), and the data were analyzed using Student’s *t*-tests for two-group comparison; data are presented as means ± standard error, and *p*-values ≤ 0.05 were considered significant.

## 3. Results

### 3.1. Differential Protein Expression and Functional Enrichment Analysis

A total of 713, 713, and 723 proteins could be detected in M, LW, and C pigs, respectively. Furthermore, we identified 135 differentially expressed proteins (DEPs) between the LW and M group, including 67 and 88 that were up- and downregulated, respectively. For C vs. M, there were 155 DEPs, of which 71 were upregulated and 64 were downregulated. For C vs. LW, there were 135 DEPs of which 71 were upregulated and 64 were downregulated (Figure 1A–C). Additionally, we used gene ontology (GO) to explore the DEP function information in three groups (Figure 1D–F) In the LW and M group, biological processes (BP), molecular function (MF), and cellular components (CC) were involved in the extracellular region, extracellular space, complement activation, and blood coagulation biological processes. Our GO analysis in the C and M group showed DEPs commonly related to the extracellular space, extracellular space, and complement activation. In the C and LW group, the DEPs are related to calcium ion binding, heparin binding, acute phase response, and complement activation, indicating they are also involved. Our results indicate that various biological changes occurred in the three pig breeds. We performed KEGG analysis to gain further insights into the biological functions of DEPs in different pig breeds (Figure 1G–I). In the LW and M group, the DEPs were enriched in disease and immune-related pathways, such as pathways for leukocyte transendothelial migration, NF-kappa B signaling, TNF signaling, and bladder cancer, among others. For the C and M group, the DEPs were mainly enriched in complement coagulation cascades, cell adhesion molecules, and prion disease systemic lupus erythematosus pathways. In the C and LW group, the DEPs mainly participated in complement coagulation cascades, *Staphylococcus aureus* infection, antigen processing, presentation, and presentation pathways.

### 3.2. Protein Expression Clustering Analysis

Proteins were classified according to their expression changes in different pig breeds through expression pattern clustering; furthermore, we speculated on the possible relationship between proteins, and specific functions were considered. Our examination of the proteins in 18 samples revealed six clusters, which we further divided into three groups according to their dynamic expression (Figure 2A,C,E and Appendix A). Group 1, including clusters 2 and 5, had opposite expression trends (Figure 2A). GO functional enrichment analysis showed that the proteins in cluster 2 were highly expressed in LW and C but lower in M pigs. Moreover, the proteins were mainly expressed in extracellular structure organization, extracellular matrix organization, zymogen activation, and other related functions. Proteins in cluster 5 were more highly expressed in M, and we observed significant enrichment of some immune-related functions (Figure 2B), including the immune system process, humoral immune response, B cell receptor signaling pathway, lymphocyte-mediated immunity, and innate immune responses. By observing the protein expression level, we confirmed that a stronger innate and adaptive immune response was observed in Min pigs than in LW and C pigs. Group 2 included clusters 3 and 4 (Figure 2C). The protein expression levels in cluster 3 were primarily highest in LW and lower in C and M pigs; however, in cluster 4, the opposite was true. In cluster 3, the primary associated GO terms were serine-type endopeptidase inhibitor activity, peptidase regulator activity pathways, as well as in protease binding pathways. The cluster 4 proteins were primarily of the categories molecular mediator of the immune response, immunoglobulin production, immune system process, and immune response (Figure 2D). Finally, clusters 1 and 6 were in group 3 (Figure 2E). The protein expression levels of clusters 1 and 6 were the highest and lowest, respectively, in C pigs. The biological process categories of proteins in cluster 1 were related to cell processes, such as cell adhesion, cellular oxidant detoxification, and cell–cell adhesion. The cluster 6 proteins were mainly involved in regulating enzyme activity, negative regulation of hydrolase activity, and negative regulation of peptidase activity in addition to the immune response (Figure 2F). Our results show that the functions of proteins highly expressed in Min pigs compared with LW and C pigs are mainly related to immunity. Furthermore, our results are consistent with previous observations, namely, that M pigs have stronger disease resistance than LW and C pigs in actual production. Generally, the disease resistance of C pigs is somewhere between that of M pigs and LW pigs; however, more evidence is required to support this.

### 3.3. Differential Gene Expression and Functional Enrichment Analysis

A total of 17,160 distinct mRNAs were detected after excluding those that were present at extremely low abundances. A total of 791 DEGs were identified in LW vs. M, in which 436 were upregulated and 355 were downregulated (Figure 3A, Appendix A). To gain a broader understanding of the biological effects of mRNA in response to complex environmental challenges, we used the GSEA tool to analyze gene enrichment pathways for all of the expressed genes. Our visualization of the top seven KEGG enriched pathways with GSEA indicates enrichment of multiple disease pathways associated with the highly expressed genes of Min pigs, such as *Staphylococcus aureus* infection, asthma, viral myocarditis, and graft-versus-host disease (Figure 3B, Appendix A). However, there was also enrichment of immune-related pathways, including cell adhesion molecules, the intestinal immune network for IgA production, and allograft rejection. Our results provide a molecular basis supporting the superiority of M pigs over LW pigs in disease resistance and immune regulation.

GSEA dot plot analysis was performed to further analyze the core genes that had a high contribution to Min pigs regarding disease and immune-related pathways. In the case of 107 genes, the cell adhesion molecule pathway (KO04514) (Appendix A) was enriched, with significantly enrichment in the case of 67 of these genes. We identified 61 genes in immune and disease-related pathways (Figure 3C). We determined that the core genes enriched in disease and immune-related pathways to be of interest because they may account for differences in disease resistance among pig breeds. We conducted KEGG interaction network analysis to explore the interaction between GSEA enrichment pathways, search for core pathways, and understand the relationship between pathways and genes at a global level (Figure 3D). The results indicate the centrality of cell adhesion molecules and antigen processing and presentation pathways. We further analyzed key genes in the interaction network and found that these genes mainly belong to MHC II (HLA-DRA, HLA-DOB, HLA-DMB, HLA-DQB1, SLA-DOA, SLA-DMA) and T cell receptors (TRBV19, TRBV16, TRAV36DV7, TRBV3, TRBV27).

### 3.4. Transcriptome and Proteome Expression Coherence Analysis

To investigate the association between protein changes and gene expression, we identified 599 significant mRNA–protein pairs between M and LW pigs (Figure 4A). A scatter plot of the nine-quadrant association analysis was then subdivided into sectors, and each was assessed for gene ontology terms. We distributed the 599 mRNA–protein pairs in nine quadrants (Figure 4B and Appendix A). Protein abundance was lower than RNA abundance in quadrants 1, 2, and 4. In contrast, in quadrants 6, 8, and 9, protein expression abundance was higher than RNA abundance, which may be caused by post-transcriptional or regulation of translation. In quadrants 3 and 7, the patterns of differentially expressed mRNAs and their corresponding proteins were consistent, indicating that changes in the transcription and translation levels were synchronous. Quadrant 5 showed no difference in protein and RNA expression. Further correlation analysis (COR = 0.0322) showed that the abundance of most differentially expressed proteins was not correlated with corresponding transcription levels, suggesting that post-transcriptional modifications may regulate pig disease resistance and innate immunity levels. The correlation re-analysis of genes and proteins in quadrants 3 and 7 quadrants showed that COR = 0.67 (Figure 4C), indicating high correlation and reliability. Gene expression in quadrants 3 and 7 was consistent at mRNA and protein levels, which warranted further attention.

GO analysis showed that the top 15 biological processes in quadrants 3 and 7 were commonly related to the blood-related pathway and immune responses, such as coagulation, negative coagulation regulation, hemostasis, humoral immune response, immune system processes, and the immune response (Figure 4D). The pathway analysis of these mRNA–protein pairs in quadrants 3 and 7 revealed annotation of six biological processes, including organismal systems, environmental processing, genetic information processing, cellular processing, metabolism, human disease, and information processing. Most genes were concentrated in organismal systems and human diseases (Figure 4E). The significantly enriched immune-related pathways, such as complement and coagulation cascades, hematopoietic cell lineage, and TNF signaling pathways, are notable. Accordingly, we also performed GO and KEGG analysis for mRNA–protein pairs in other quadrants, including quadrants 2 and 8 (Appendix A), 4 and 6 (Appendix A), and 1 and 9 (Appendix A).

### 3.5. Integration Analysis of Transcriptomics and Proteomics

A total of 16 mRNA–protein pairs (glutathione peroxidase (GPX3), lipopolysaccharide-binding protein (LBP), ENSSSCG00000031037 (Ig-like domain-containing protein), cadherin 2 (CDH2), insulin-like growth factor 1(IGF-1), neural EGFL-like 2 (NELL2), collagen type XII alpha 1 chain (COL12A1), dermatopontin (DPT), vascular cell adhesion molecule 1 (VCAM1), CD14 molecule (CD14), lactate dehydrogenase B (LDHB), serpin family B member 1 (SERPINB1), ENSSSCG00000030830, ESSCG00000033114, ENSSSCG00000035256, and ENSSSCG00000031595) were found to differ between the Min and LW pigs at both mRNA and protein levels (Figure 5A). Five of the genes are new, and the other eleven have been previously reported. Most of the expression of mRNA–protein pairs presented the same trend of either being upregulated or downregulated (NELL2, CDH2, COL12A1, IGF1, CD14, LBP, SERPINB1, ENSSSCG0000001037, ENSSSCG00000030830, and ENSSSCG00000033114). Six genes had inconsistent expression trends at the protein and mRNA levels (GPX3, VCAM1, DPT, LDHB, ENSSSCG00000035256, and ENSSSCG00000031595) (Figure 5B,C). The pathway enrichment analysis of these 16 mRNA–protein pairs revealed several immune-related clusters, such as the NF-kappa B signaling pathway, cell adhesion molecules (CAMs), Toll-like receptor signaling pathway, and the HIF-1 signaling pathway (Table 1). Many proteins were intimately involved in these pathways (VCAM1, CD14, LDHP, CDH2, IGF1, SERPINB1, and LBP), which is promising for management of abnormal diseases (Table 1).

### 3.6. Co-Location Analysis of 16 Key DEGs, DEPs, and Pig QTLdb

Integrated immune and disease-related traits from the pig QTLdb database were detected either by QTL mapping or genome-wide association studies (GWAS), comparing their chromosome positions to gain further insight into the association between DEGs and immune traits. We located 13 DEGs and DEPs (NELL2, CDH2, COL12A1, IGF1, VCAM1, LDHB, CD14, LBP, SERPINB1, GPX3, ENSSSCG00000031595, ENSSSCG00000031037, and ENSSSCG00000030830) within or overlapped with immune-related QTL regions (Figure 6). These QTLs were mainly located on chromosomes 1, 2, 4, 5, 7, 14, 16, and 17.

### 3.7. Correlation Analysis of 16 DEPs and Cytokines

To further verify differences in the immunity, we measured the abundance of 10 inflammatory cytokines in Min and LW pig serum. Our results reveal that TNF-α, IL-6, IL-10, and GS-CSF protein concentrations were higher in the LW pigs (Figure 7A–D). Correlation analysis showed that NELL2, COL12A1, LDHB, and ENSSSCG00000031595 were significantly positively correlated with the expression of anti-inflammatory interleukin (IL) 10, and pro-inflammatory cytokine IL6 and tumor necrosis factor alpha (TNF-α), which implies a potential regulatory function (Figure 7E).

### 3.8. PRM Validation of 16 DEPs

According to our transcriptome and proteomics analyses, whether alone or a combination of both, we found that many molecules associated with immunity and inflammation were significantly expressed in different pig breeds. To further validate these results, we selected some critical candidate proteins, including CD14, SERPINB1, VCAM1, NELL2, and LDHB, and performed PRM to validate the expression levels of these proteins (Figure 7F). These results are basically consistent with those of DIA, indicating the reliability and reproducibility of the DIA-derived proteomics results in our study.

## 4. Discussion

It is of great significance to explore the disease resistance mechanism of Chinese local and foreign pig breeds for improving their health and productivity. In this study, we integrated results from proteome and transcriptome analysis to investigate the immune mechanisms operating in different pig breeds and determine a possible molecular rationale for the differences in disease resistance among pig breeds. As we know, both the body weights and the days of age are important to the transcriptomic and proteomic analysis. Compared to the weight, the days of age maybe more suitable to present the immune system growth period for animals, so we chose same age instead of same weight of animals for analysis. However, weights differences between the M and LW pigs cannot be ignored.

We analyzed three pig breeds and identified a total of 723 proteins in serum that were similar to the peripheral blood serum of piglets (624 proteins) [15], which is different from that of the pig lymph nodes (6296 proteins) of LW and Tongcheng pigs [16] and tenderloin (1448 proteins) of Tibetan and Yorkshire pigs [17], possibly as a consequence of the tissue-specific expression of proteins. In our study, cluster expression analysis showed that the cluster proteins with high specific expression in Min pigs were mainly enriched in the immune system process as well as the humoral immune and innate immune responses in addition to other immune-related GO terms. However, in LW and C pigs, the proteins of the high expression cluster were mainly associated with peptidase-regulator and intracellular activity terms, which suggests differences in the immune function of different breeds of pigs.

GSEA analysis showed that the cell adhesion molecule and antigen processing and presentation pathways were significantly enriched in M pigs. These results are similar to studies of Tongcheng pigs and LW pigs regarding PRRSV [14], indicating that the described pathways may play crucial roles in Chinese local pig disease resistance. Moreover, the key genes in the interaction network mainly belong to the categories of MHC II and T cell receptors. The results of previously reported studies indicated that differences in the ability of MHC molecules to bind and present antigenic peptides resulted in differences in the intensity of the response to antigens in different individuals [18]. The MHC was found to be highly polymorphic within the population, with different individuals carrying different MHC alleles and encoding different amino acid sequences of MHC molecules, resulting in differences in the abilities of different MHC genes to bind and present antigenic peptides [19,20]. Our previous studies showed that the number of nucleic and aminic mutations located at the antigen-binding site of SLA class I genes/proteins was greater in the Min pig than in the LW pigs [6]. MHC gene polymorphisms enable genetic regulation of the immune response at the population level and confer great adaptability and resilience to species. However, it should be noted that although MHC II and T cell receptor-related genes were significantly enriched in Min pigs, these genes were not found to be differentially expressed between the studied breeds. This may be related to the fact that Min pigs were in a relatively healthy state and were not disturbed by pathogenic microorganisms. The results of GSEA further confirmed that more immune-related genes were significantly enriched in Min pigs than in LW pigs at the transcriptional level. It is reasonable to speculate that the differential expression of these genes enhances the disease resistance of Min pigs when the challenge occurs.

In our data, 16 differentially co-expressed genes were identified at the transcriptional and protein levels, and 10 of these showed the same expression trend at both transcriptional and protein levels. Six proteins, namely, *NELL2, IGF1, COL12A1, LBP*, and *ENSSSCG00000031037*, were significantly upregulated in LW pigs, indicating continuous synthesis and secretion of immune molecules in response to antigenic stimuli. These six genes have been extensively studied in the context of human diseases. *NELL2* is a neuron-specific secretory protein widely involved in various disease processes [21,22], and interference with NELL2 inhibits the interferon-beta (IFN-β) response [21]. The *COL12A1* gene is associated with multiple disease prognoses and immune infiltration, and high *COL12A1* expression in cancer tissues can promote disease development and progression [23,24]. IGF-1 is a growth factor that plays a critical role in cell proliferation and body growth and is positively correlated with the development of several specific diseases and cancers [25,26]. As part of a self-regulatory mechanism to prevent hyperimmunity, plasma LBP is significantly increased during the acute phase of inflammation, and high concentrations of LBP could inhibit the release of LPS-mediated inflammatory cytokines, thereby inhibiting the activation of cell signaling. As a novel gene, reports of ENSSSCG00000031037 are rare, though it is found to be highly expressed in the porcine defect disease scrotal hernia [27]. CDH2 is a membrane immune protein that mediates cell–cell adhesion. CDH2 promotes malignancy by stimulating FGFR. The overexpression of CDH2 on tumor cells can mediate invasion and metastasis [28].

The CD14, SERPINB1, ENSSSCG00000030830, and ESSCG00000033114 proteins were significantly upregulated and expressed in Min pigs. As a monocyte differentiation antigen, CD14 can regulate innate immunity [29]. CD14 is associated with many diseases and plays a dual role in host resistance to infection. High expression of CD14 can benefit the host by promoting inflammation and immune responses that eliminate invading pathogenic microorganisms. However, it can also adversely affect the host due to excessive inflammation and the spread of pathogens [30]. SERPINB1, previously called MNEI (monocyte/neutrophil elastase inhibitor), is mainly found in the cytoplasm of neutrophils and monocytes [31]. The results of previous research reported high levels of SERPINB1 in myeloid cells, especially neutrophils, the short-lived cells recruited immediately to infection sites to destroy pathogenic microbes [32]. In bacterial lung infection, SERPINB1 protects against inflammatory tissue injury and neutrophil death [33]. In mice, the results of SERPINB1^−/−^ studies demonstrated that SERPINB1 protects pulmonary antimicrobial proteins from proteolysis during microbial infection [33]. In SERPINB1^−/−^, there was insufficient recruitment of neutrophils to the lungs, and mice could not effectively remove bacteria [32,33].

The consistency between mRNA and protein can verify the reliability of sequencing data to a certain extent, while differences between the two often indicates post-transcriptional interference [34]. Furthermore, we observed six genes with inconsistent expression trends at protein and mRNA levels (*GPX3*, *VCAM1*, *DPT*, *LDHB*, *ENSSSCG00000035256*, and *ENSSSCG00000031595*) in M and LW pigs. This suggests the possibility of post-transcriptional regulation, epigenetic modification (DNA methylation, histone modification, etc.), or post-translational regulation occurring after mRNA formation. Transcriptomic or proteomic data usually only reflect the net effect of the regulatory system and the catabolic equilibrium state. The inconsistency between the two is only a reflection of the alternation between synthesis and degradation processes [34,35]. VCAM, a vascular endothelial and stromal adhesion molecule, is an immune activation regulator (immune checkpoint). VCAM1 protects hematopoietic stem cells (HSCs) and leukemic stem cells (LSCs) from phagocytosis through haploidentical monocytes [32]. VCAM1^−/−^ mice showed reduced numbers of hematopoietic stem and progenitor cells (HSPCs) in the spleen [36,37]. GPx plays a crucial role in eliminating all forms of hydrogen peroxide produced in the body [38]. GPX3 in plasma and on mucosal surfaces protects epithelial cells from oxidative damage. In addition to detoxification, GPX3 has a significant role in cellular defense mechanisms, such as inflammation and cancer progression inhibition. GPX3 has been proved to have a significant tumor-suppressive effect in various cancers [39,40]. In addition, the presence of GPX3 helps eliminate inflammation in the tumor microenvironment [41,42]. DPT is a matricellular protein with cardinal roles in cutaneous wound healing [43]. Human lactate dehydrogenase (LDH), consisting of two subunits, LDHA and LDHB, catalyzes the interconversion of pyruvate and lactate in the anaerobic glycolytic pathway and is a key glycolytic enzyme. LDHB participates in host cell mitochondrial metabolism and innate immune response through the NFKB signaling pathway. LDHB inhibition promotes CSFV growth via mitophagy, whereas its overexpression results in decreased CSFV replication [44]. In summary, most of these significantly upregulated genes/proteins in LW pigs have been demonstrated to promote the development of diseases in mammals to varying degrees. The genes/proteins with high expression in Min pigs are mostly related to inflammation and disease inhibition. However, direct evidence of this in pig studies is still lacking.

Pro-inflammatory cytokines interleukin-6 (IL-6) and tumor necrosis factor-α (TNF-α) and anti-inflammatory cytokine interleukin-10 (IL-10) were found to be significantly upregulated in LW pigs, which mediated the inflammatory response. TNFα is a pleiotropic pro-inflammatory cytokine. The primary function of TNFα is the upregulation of multiple pro-inflammatory proteins (chemokines, cytokines, adhesion molecules, growth factors, etc.) by activation of transcription factor nuclear factor κB (NF-κB) and mitogen-activated protein kinase (MAPK) pathways [45]. TNF-α can not only inhibit viral replication but also be inhibited by viral particles. IL-6 is a pleiotropic cytokine produced mainly by monocytes/macrophages and epithelial cells. IL-6 regulates the physiological functions of multiple immune and non-immune cell types and represents critical interphase between immune, endocrine, and neural systems. IL-6 affects adaptive immune responses by stimulating the differentiation of T cells and B cells and promoting immunoglobulin production [46,47]. IL-10, a cytokine secreted by lymphocytes, plays an important role in the development of immunosuppression by promoting humoral immunity and inhibiting cellular immunity and inflammatory response [46]. Some studies have speculated that IL10 is secreted in large quantities at the early stage of PRRSV infection, resulting in ineffective viral clearance, which may be one of the strategies for PRRSV to suppress the host immune response [47]. Correlation analysis showed that NELL2, COL12A1, LDHB, and ENSSSCG00000031595 proteins were significantly positive correlated with the expression of IL-6, TNF-α, and IL-10. This result suggests that differential expression of these proteins may regulate the secretion of cytokines by immune cells, which are involved in the body’s innate and adaptive immune responses to maintain homeostasis. Moreover, further research such as the analysis of SNPs, CNVs and indels on these genes should be done to study the affection mechanism of these genes.

## 5. Conclusions

In conclusion, we investigated the transcriptomics and proteomics differences in the immunity between resistant and susceptible pig breeds. The differentially expressed genes and proteins in M pigs demonstrated significant function enrichment in immune-related antigen processing and presentation and cell adhesion pathway than LW pigs. In addition, we found 16 genes associated with immunity and disease that were differentially co-expressed at the mRNA and protein levels. Some genes/proteins, such as NELL2 and LDHB, are potential critical targets for immune responses, and the data from this study are expected to be helpful for the future breeding of disease-resistant pigs.

## Figures and Tables

**Figure 1 biology-11-01708-f001:**
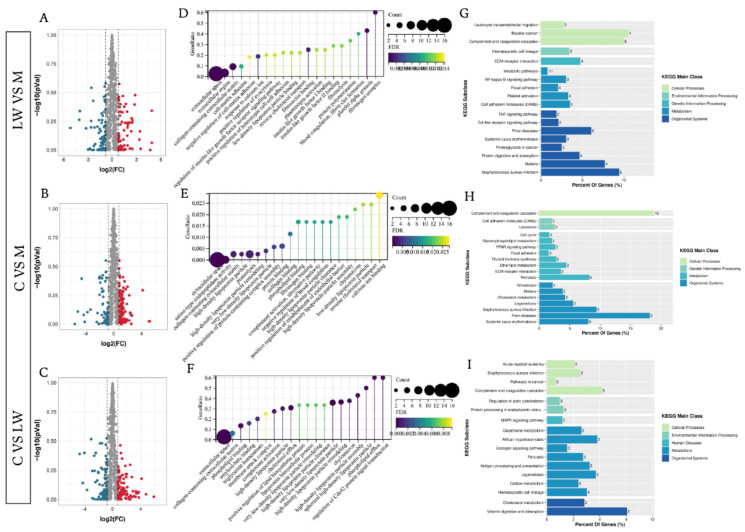
Visualization of proteomic results between different pig breeds. (**A**–**C**) Volcano plots of comparation for Large white (LW) with Min pig (M), M with cross-bred of LW×M (**C**), and LW with C. Significantly differentially expressed proteins are highlighted in red and green, and grey dots represent non-differentially expressed proteins. (**D**–**F**) Enrichment analysis using gene ontology (GO) for LW with M, M with C, and LW with C. (**G**–**I**) Bar diagram of partial significantly enriched pathways for LW with M, M with C, and LW with C DE proteins.

**Figure 2 biology-11-01708-f002:**
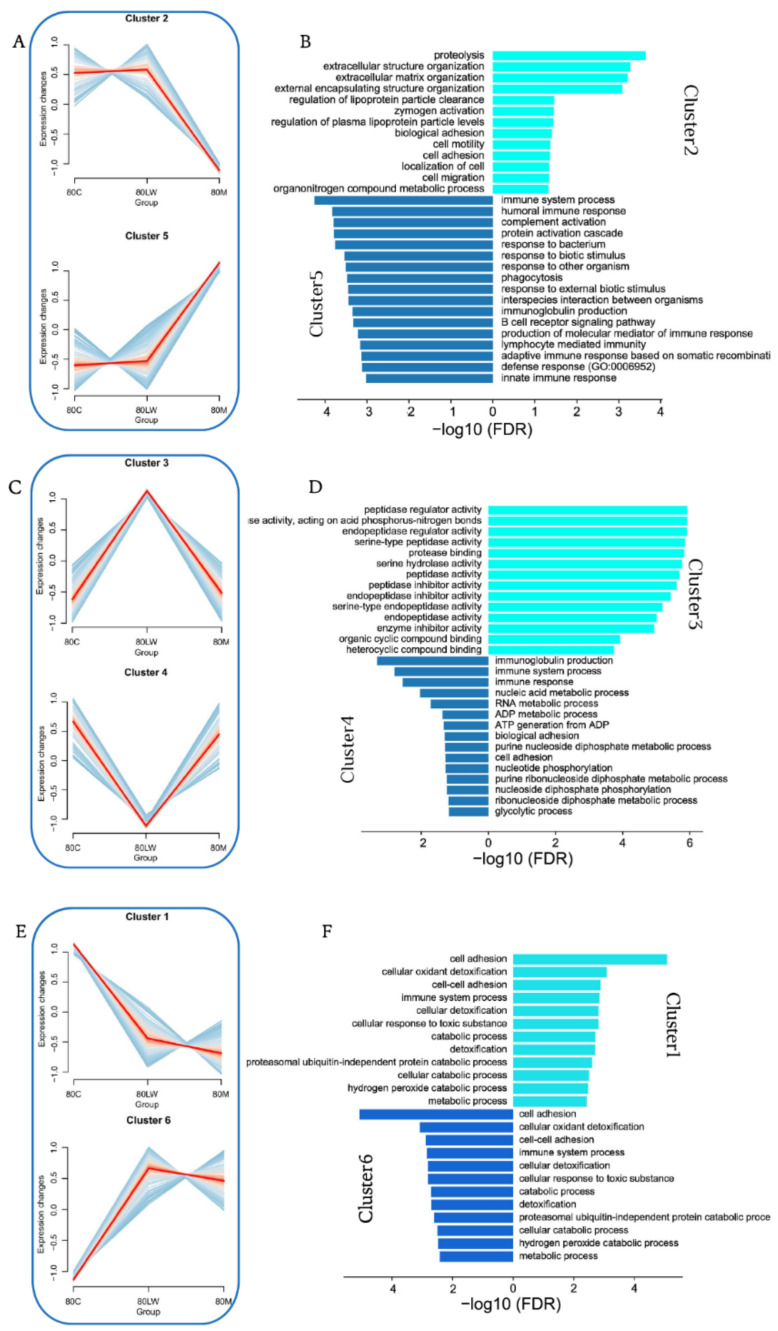
Protein cluster expression and function analysis. (**A**,**C**,**E**) Cluster analysis of all proteins based on expression. All the proteins clustered into six clusters, clusters with opposite expression trends were classified into one group, and distinct clusters were assigned to different breed pigs according to the expression patterns. (**B**,**D**,**F**) Analysis of the functions of proteins in the different groups. The bar plot shows significant terms by gradient legend as *p*-value < 0.05. The *x*-axis corresponds to −log10 (FDR) proteins belonging to the given enriched terms. Different colors represent different clusters in a same group.

**Figure 3 biology-11-01708-f003:**
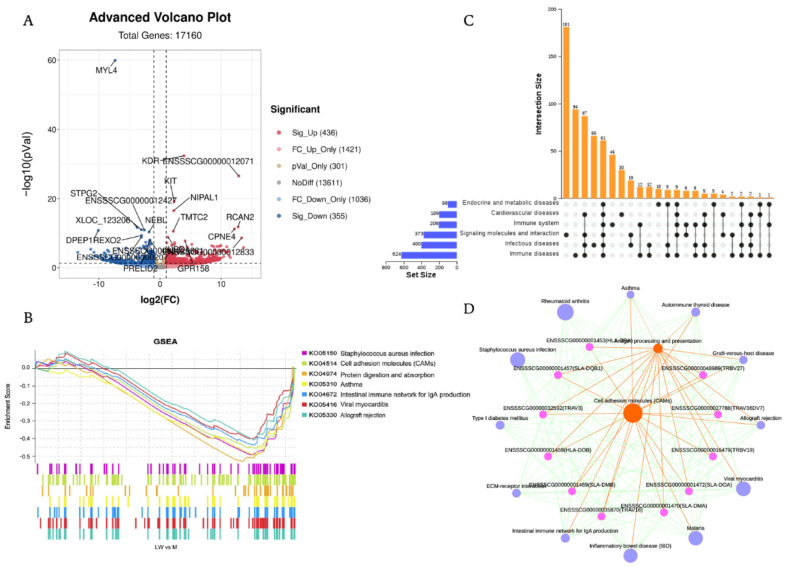
Visualization of transcriptomics results of mRNAs between M and LW in blood serum. (**A**) Volcano plot of all mRNA alterations between M and LW pigs. The horizontal dashed line corresponds to *p*-value < 0.05, and the vertical dashed line corresponds to a 1−log2 (fold change) (decrease or increase) in expression levels. Red, blue, and gray dots represent upregulated, downregulated, and non-differentially expressed genes in LW pigs (as compared with M pigs). (**B**) GSEA plots partially reveal key pathways within M and LW pigs. Different color squares represent different pathways. *p*-value ≤ 0.05 and q-value ≤ 0.25 were considered significant. (**C**) Core co-expressed genes in immune and disease-related pathways. (**D**) GSEA significant enrichment pathway and related gene interaction network diagram. The number of genes enriched in the pathway was used to change node size.

**Figure 4 biology-11-01708-f004:**
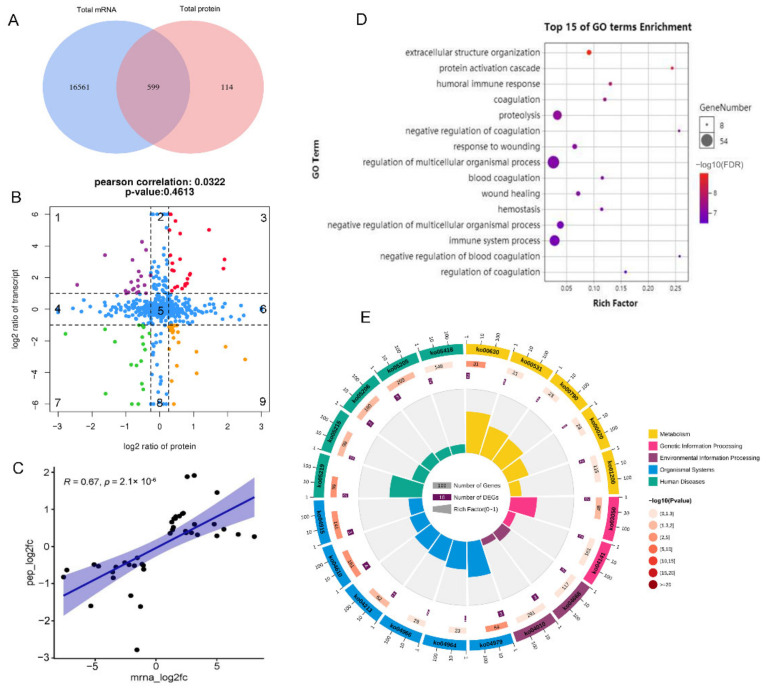
Expression concordance of proteotranscriptomic results between M and LW pigs. (**A**) Venn diagram of the gene set identified both in transcriptome and proteome. (**B**) Nine-quadrant diagram of mRNA-proteins pairs. (**C**) Gene correlation analysis in quadrants 3 and 7. (**D**) Bubble diagram of partial biology progress GO term clusters of quadrants 3 and 7. (**E**) Bar diagram of significantly enriched pathways for quadrants 3 and 7. The mRNA-proteins pairs with log2 ratio in quadrant 1 were represent in purple, in quadrant 3 were represent in red, in quadrant 7 were represent in green, in quadrant 9 were represent in orange, and in quadrant 2, 4, 5, 6 were represent in blue.

**Figure 5 biology-11-01708-f005:**
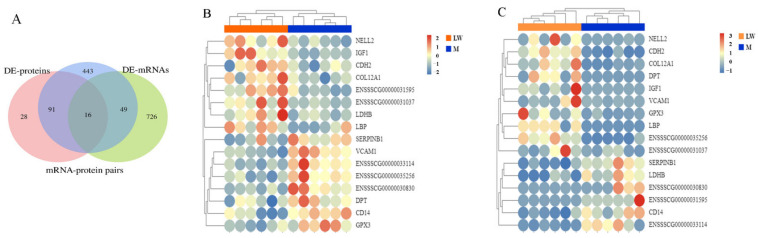
Expression concordance of proteotranscriptomic results for blood serum between M and LW pigs. (**A**) Venn diagram of gene sets of mRNA–protein pairs, DE mRNAs, and DE proteins. (**B**,**C**) Heatmaps of expression of 16 overlapping mRNAs.

**Figure 6 biology-11-01708-f006:**
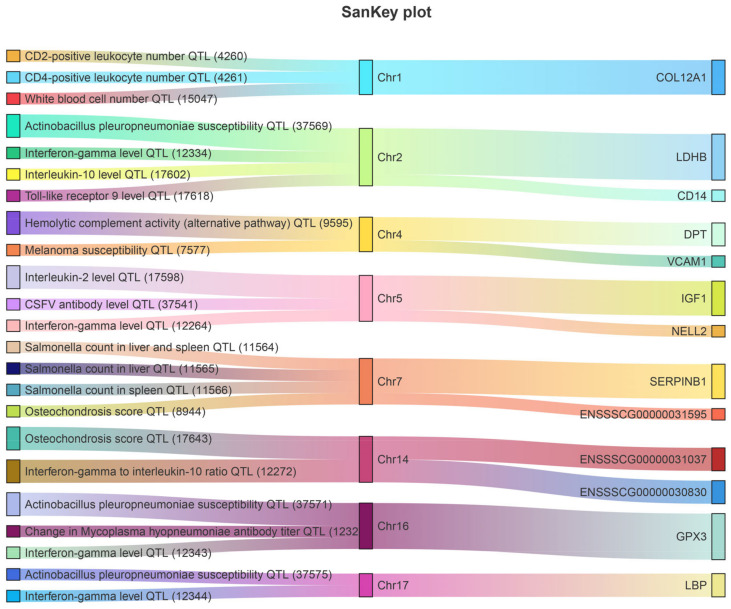
The intersection of 16 key genes with swine immunity and disease-related QTL data.

**Figure 7 biology-11-01708-f007:**
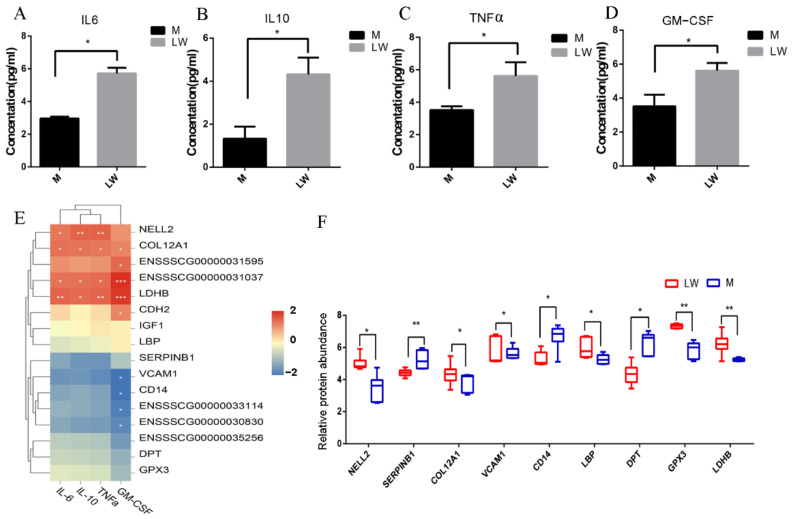
Quantification of cytokine and protein expression abundance. (**A**–**D**) Quantification validation of cytokines. (**E**) Correlation matrix of the expression levels of the proteins and cytokines. (**F**) Quantification validation of proteins by PRM. *: *p*-value < 0.05; **: *p*-value < 0.01.

**Table 1 biology-11-01708-t001:** KOBAS pathway enrichment result.

Pathway	ID	Q-Value	Gene
NF-kappa B signaling pathway	ko04064	0.000422632	*VCAM1*, *CD14*, *LBP*
Cell adhesion molecules (CAMs)	ko04514	0.002354882	*CDH2*, *VCAM1*, *ENSSSCG00000035256*
Toll-like receptor signaling pathway	ko04620	0.004674842	*CD14*, *LBP*
HIF-1 signaling pathway	ko04066	0.006094204	*IGF1*, *LDHB*
Transcriptional misregulation in cancers	ko05202	0.02013848	*CD14*, *IGF1*
Tuberculosis	ko05152	0.02894842	*CD14*, *LBP*
MAPK signaling pathway	ko04010	0.03395784	*CD14*, *IGF1*
Propanoate metabolism	ko00640	0.03442503	*LDHB*
Aldosterone-regulated sodium reabsorption	ko04960	0.03641682	*IGF1*
Pyruvate metabolism	ko00620	0.03939765	*LDHB*
Ovarian steroidogenesis	ko04913	0.04632108	*IGF1*
African trypanosomiasis	ko05143	0.04730651	*VCAM1*
Cysteine and methionine metabolism	ko00270	0.04927466	*LDHB*

Note: NELL2 (Neural EGFL Like 2), IGF1 (Insulin-Like Growth Factor I), LBP (Lipopolysac cha ride binding protein), VCAM1 (Vascular Cell Adhesion Protein 1), LDHB (L-Lactate Dehydrogenase B Chain).

## Data Availability

The datasets presented in this study can be found in online repositories. The names of the repositories and accession number (s) can be found in Section 2.

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
