# Peer review of "Integrated Proteotranscriptomics Reveals Differences in Molecular Immunity between Min and Large White Pig Breeds"

_biology, 2022, doi:10.3390/biology11121708_

Round 1
Reviewer 1 Report
It’s important to reveal molecular differences on immunity between Min and Large White pig breeds. The manuscript is well written and the data are interesting according to my opinion, even though some issues are to be better argued.
Line 4, Both numbers and symbols should be superscript.
Line 23, The experimental design was not clearly described in the abstract. I suggest authors to give a brief statement.
Line 91, Please clearly state out the crossbreeding of the F1 (M×LW or LW×M?).
Line 93, As far as we know, the growth rate of M and LW are different. Is it possible to select piglets with the similar weights even the pigs were fed under the same condition? And, how about the average weight (kg) of the selected piglets? The herein information was missing.
Line 95, We actually stop the nursery on the 8th week (about 56-60 days of age), why did the author say it was 80th day? Again, what’s the biology meaning of this time point? Also, I can not get the time information on sample collections, because you described as “after which…” (I can get the info. in Line 139).
Line 139, What kinds of blood cells did the author use for transcriptome experiment? Whole white blood cells or PBMC?
Line 150, I suggest to firstly use the abbreviates of C, M, LW in the part of 2.1, and, keep the consistence of “C” and “F1”.
Line 152, Here and elsewhere throughout the manuscript, the writings of “p value” are not consistent. For example, the authors used “p-value”, “P”, “p”, “p”…
Line 162, What is the meaning of “C2”?
Line 219, Here and elsewhere, please note the spaces. (…they are…).
Line 230, I suggest to update the figure with higher resolution.
Line 232, “black dots” should be “gray dots”?
Line 266, please add the info. on “previous observations” in brief.
Line 268, human pigs?
Line 279-280, I suggest to provide the supplemental data on the annotated DEGs.
Line 282, It seems that the exact DEGs used for GSEA enrichments were not clearly statement, so it may be a little bit of difficult to understand the results here.
Line 283-289, Because the authors did the enrichments based on DEGs of LW vs M, it is hard to say that the M pig has the superiority on resistance. The descriptions here might need a fine organization.
Line 291, Please move the figure 3 after the next paragraph, since figures should be attached upon it has been already described.
Line 314, The authors stated that transcriptions of blood cells and proteins from sera were compared. My question is, why not comparing transcriptions and proteins from blood cells?
Line 316, Did the author mean that both genes and proteins are DEGs and DEPs in their respective experiment for “mRNA–protein pairs”?
Line 325 and 329, digit bits of COR are not consistent.
Line 333, “Differentially Expressed Genes” in Figure 4A; quadrant numbers should be marked in 4B; please move the figure after the next paragraph.
Line 387, the numbers in brackets should be indicated/illustrated in figure legend.
Line 397, please move the figure after the next paragraph.
Line 404, The authors stated that some critical candidate proteins/genes were discovered between M and LW breeds, how about the mutations, such as the SNP, on these genes between the two breeds?
Line 406, Did the author mean that the increased expression of proteins are results of LW vs M? While, as shown in the Figure 7F, it seems that other proteins such as NELL2 also have the increased expressions except for GPX3. Descriptions here is somewhat baffling.
Line 417, “(624 proteins)”
Line 427, gene info. that used for GSEA are not clearly described.
Line 438-440, please rewrite the sentences here.
Line 457-458, 467-468, and 506-509, Because expressions of NELL2 and LBP are higher in LW, and GPX3 is lower in M, is that mean LW is more resistant than M? Could we get the conclusion based on the simple comparations?
Line 502, “VCAM1–/–” mice.
Line 548-549, Conclusions should be optimized.
Author Response
It’s important to reveal molecular differences on immunity between Min and Large White pig breeds. The manuscript is well written and the data are interesting according to my opinion, even though some issues are to be better argued.
- Line 4, Both numbers and symbols should be superscript.
Response:Done. (Lines 4-5)
- Line 23, The experimental design was not clearly described in the abstract. I suggest authors to give a brief statement.
Response:Follow your suggestion, we have added the number and the raising environment of the animals we used in the experiment. (Lines 26-27)
- Line 91, Please clearly state out the crossbreeding of the F1 (M×LW or LW×M?).
Response:Thank you for your precious comments and advice. In our study, we used the cross-bed of LW×M F1 pigs, and corresponding description has been added (Line 99).
- Line 93, As far as we know, the growth rate of M and LW are different. Is it possible to select piglets with the similar weights even the pigs were fed under the same condition? And, how about the average weight (kg) of the selected piglets? The herein information was missing.
Response:Thanks for your suggestion. As we know, both the weights and the days of age are important when we analysis the transcriptomic and proteomic differences between pigs. Compared to the weight, we think the days of age is more suitable to present the immune system growth period for animals, so we chose same age animals for analysis. However, weights differences between the M and LW pigs can not be ignored. And we have added the average weight of the selected piglets in the M&M section (Lines 102-103) and the potential influence of weight in discussion (Lines 440-445).
- Line 95, We actually stop the nursery on the 8th week (about 56-60 days of age), why did the author say it was 80th day? Again, what’s the biology meaning of this time point? Also, I can not get the time information on sample collections, because you described as “after which…” (I can get the info. in Line 139).
Response:Thank you for your comment. In our farm, we raised the piglets for 35 days in the farrowing house and 45 days in the nursery house. And that is the main reason we chose 80th day. And moreover, in previous studies (Sieverding et al., 2000), the immune status was stable after 70th day, and the 80th day is more suitable for our research purposes. We have added the raising procedure in the M&M section (Line 102-103).
Sieverding, E.: Handbuch gesunde Schweine; Kamlage Verlag (2000).
- Line 139, What kinds of blood cells did the author use for transcriptome experiment? Whole white blood cells or PBMC?
Response:In our study, we used whole white blood cells (WBC) for transcriptome experiment. Some research showed that compare to PBMC (lymphocytes and monocytes), WBC is more suitable for generating gene expression data with minimal variability and higher sensitivity(1). Although the cells detected in PBMC are only a small part of the total WBC, the expression of specific genes in PBMC can still be detected in WBC (2). The PBMC and WBC expression profile correlation was between 0.78 and 0.91 (3, 4). We have added the blood cells information in the M&M section (Lines 154-155)
- He D, Yang CX, Sahin B, Singh A, Shannon CP, Oliveria J-P, et al. Whole Blood Vs Pbmc: Compartmental Differences in Gene Expression Profiling Exemplified in Asthma. Allergy, Asthma & Clinical Immunology (2019) 15(1):67.
- Colette D, Elodie M, Tatiana Z, Bertrand Bh, Sonja H, Aurélie, et al. Transcriptomes of Whole Blood and Pbmc in Chickens. Comparative Biochemistry and Physiology Part D: Genomics and Proteomics (2016) 20:1-9.
- Min JL, Barrett A, Watts T, Pettersson FH, Lockstone HE, Lindgren CM, et al. Variability of Gene Expression Profiles in Human Blood and Lymphoblastoid Cell Lines. BMC Genomics (2010) 11:96. Epub 2010/02/10.
- Bondar G, Cadeiras M, Wisniewski N, Maque J, Chittoor J, Chang E, et al. Comparison of Whole Blood and Peripheral Blood Mononuclear Cell Gene Expression for Evaluation of the Perioperative Inflammatory Response in Patients with Advanced Heart Failure. PLoS One (2014) 9(12):e115097. Epub 2014/12/18.
- Line 150, I suggest to firstly use the abbreviates of C, M, LW in the part of 2.1, and, keep the consistence of “C” and “F1”.
Response:Thank you for the suggestion. We have added the full name C, M and LW in the article (Line 63 and Line 99) and revised all "F1" to "C" in our full manuscript.
- Line 152, Here and elsewhere throughout the manuscript, the writings of “p value” are not consistent. For example, the authors used “p-value”, “P”, “p”, “p”…
Response:Done. (Lines 187-188, Lines 214-215, Line 293, Line 327, Line 331).
- Line 162, What is the meaning of “C2”?
Response:The gene sets in the Molecular Signatures Database (MSigDB) are divided into 9 major collections, and KEGG belongs to the C2 collection. (Line 175)
- Line 219, Here and elsewhere, please note the spaces. (…they are…).
Response:Done. (Lines 234).
- Line 230, I suggest to update the figure with higher resolution.
Response:Done. (Lines 245-246)
- Line 232, “black dots” should be “gray dots”?
Response:Done. (Line 249)
- Line 266, please add the info. on “previous observations” in brief.
Response: Follow your kindly suggestion, we have added the information about “previous observations” in the introduction. (Lines 77-80).
- Line 268, human pigs?
Response:The mistake has been revised. (Line 285).
- Line 279-280, I suggest to provide the supplemental data on the annotated DEGs.
Response:Done. (Supplemental Table S3)
- Line 282, It seems that the exact DEGs used for GSEA enrichments were not clearly statement, so it may be a little bit of difficult to understand the results here.
Response:According to the instructions for GSEA analysis, GSEA analysis is performed using all genes, not DEGs. The default input of GSEA software is the gene set (gene expression matrix and sample grouping), which can directly use the expression of all genes for analysis without differential analysis, and can detect the gene set that is not significant but has the same differential expression trend. Brief descriptions have been added in the manuscript (line 301).
- Line 283-289, Because the authors did the enrichments based on DEGs of LW vs M, it is hard to say that the M pig has the superiority on resistance. The descriptions here might need a fine organization.
Response:Thank you for your comment. We agree with you that KEGG and GO enrichment analysis based on DEGs can not distinguish whether the DEGs are up-regulated or down-regulated in the pathway. This is because the DEGs enrichment analysis does not consider the variation trend of gene expression. But in the analysis of GSEA, according to the value of the differential fold, the descending order was used to show the change trend of gene expression between the two groups. The top of the sorted gene list can be regarded as up-regulated differential genes (LW vs M), and the bottom is down-regulated differential genes (LW vs M). From our GSEA results, we can see that many immune or disease-related genes are clustered at the bottom, so those genes were significantly enriched in Min pigs. We have added brief descriptions in the manuscript (line 303).
- Line 291, Please move the figure 3 after the next paragraph, since figures should be attached upon it has been already described.
Response:Done. (Lines 323-324)
- Line 314, The authors stated that transcriptions of blood cells and proteins from sera were compared. My question is, why not comparing transcriptions and proteins from blood cells?
Response:Proteins in serum are very important, which are closely related to almost all organs, tissues and cells, and can directly reflect the pathological and physiological state of the body from the nature of proteins. Most of the markers related to diseases are related to proteins in blood, so we used serum to detect proteome and cytokines. Peripheral blood contains abundant subsets of immune cells, such as granulocytes, eosinophils, neutrophils, monocytes, granulocytes, dendritic cells and so on. When animals are challenged with disease, these immune cells secrete immune factors into the serum. Therefore, we chose whole blood for transcriptome sequencing.
- Line 316, Did the author mean that both genes and proteins are DEGs and DEPs in their respective experiment for “mRNA–protein pairs”?
Response:In our study, we used all of the expressed genes and proteins for matching mRNA–protein pairs. Brief descriptions have been added in results section (line 301).
- Line 325 and 329, digit bits of COR are not consistent.
Response:Thank you for your careful review. The mistake has been correct. (Figure 4C).
- Line 333, “Differentially Expressed Genes” in Figure 4A; quadrant numbers should be marked in 4B; please move the figure after the next paragraph.
Response:Follow your suggestion, we have revised Figure 4A and Figure 4B, and move the figure to the end of next paragraph.
- Line 387, the numbers in brackets should be indicated/illustrated in figure legend.
Response:Done. (Figure 4B)
- Line 397, please move the figure after the next paragraph.
Response:Done. (Lines 430-431)
- Line 404, The authors stated that some critical candidate proteins/genes were discovered between M and LW breeds, how about the mutations, such as the SNP, on these genes between the two breeds?
Response:Thank you for your comment. We did not sequenced the genome of the individuals, so we did not have SNPs data of these candidate genes. Actually, in our further research plan the SNPs, CNVs and Indels on these genes will all be deeply analysis. And we have added this in the discussion section (Lines 568-570).
- Line 406, Did the author mean that the increased expression of proteins are results of LW vs M? While, as shown in the Figure 7F, it seems that other proteins such as NELL2 also have the increased expressions except for GPX3. Descriptions here is somewhat baffling.
Response:We apologize for not describing it more clearly and we think this sentence is useless, so we deleted it. (Line 428)
- Line 417, “(624 proteins)”
Response:Done. (Line 447)
- Line 427, gene info. that used for GSEA are not clearly described.
Response:In our study, we used all of the expressed genes and proteins for matching mRNA–protein pairs. Brief descriptions have been added in results section (line 301).
- Line 438-440, please rewrite the sentences here.
Response:We agree with the comment and re-wrote the sentence has been re-wrote as follow: Our previous studies showed that the number of nucleic and aminic mutations located at the antigen-binding site of SLA class I genes/proteins was greater in the Min pig than in the LW pigs. (Lines 468-470)
- Line 457-458, 467-468, and 506-509, Because expressions of NELL2 and LBP are higher in LW, and GPX3 is lower in M, is that mean LW is more resistant than M? Could we get the conclusion based on the simple comparations?
Response:Thanks for your comment. Our conclusion that M is more resistant than LW is based on a three-part analysis. 1. Cluster analysis of protein showed that the proteins higher expressed in Min pig were mainly related to innate or adaptive immunity; 2. GSEA analysis showed that immune-related pathways or genes such as antigen processing and presentation, cell adhesion molecules and MHC class II molecules were significantly enriched in Min pigs; 3. We reviewed a large number of literatures about the 16 genes identified by integrated transcription-proteome analysis and found that the proteins highly expressed in LW pigs were mainly related to pro-inflammation or promoting disease development, while the proteins highly expressed in Min pigs were mostly related to anti-inflammation or inhibiting disease occurrence and development. However, this conclusion needs to be verified. In addition, cytokine results showed that pro-inflammatory cytokines were highly expressed in large white pigs. In conclusion, we believe that these molecular differences may be partly the evidence to explain the differences in disease resistance between Chinese and foreign pig breeds. But all the conclusions should be proved by solid experiment results. And we have optimized the conclusion to avoid absolute conclusion. (Lines 572-579)
- Line 502, “VCAM1–/–” mice.
Response:Done. (Line 528)
- Line 548-549, Conclusions should be optimized.
Response:We gratefully appreciate for your valuable suggestion and we have optimized the conclusions to: “In conclusion, we investigated the transcriptomics and proteomics differences in the immunity between resistant and susceptible pig breeds. The differentially expressed genes and proteins in M pigs demonstrated significant function enrichment in immune-related antigen processing and presentation and cell adhesion pathway than LW pigs. In addition, we found 16 genes associated with immunity and disease that were differentially co-expressed at the mRNA and protein levels. Some genes/proteins, such as NELL2 and LDHB, are potential critical targets for immune responses, and the data from this study are expected to be helpful for the future breeding of disease-resistant pigs. (Lines 572-579).
Reviewer 2 Report
In this study, the authors used transcriptomics and proteomics analysis to analyze the immune differences between a resistant (Min pig) and a susceptible (Large White, LW) pig breed. The topic is interesting, and the experiments were well designed. The the methods are used appropriately. I recommend for accept after some minor revisions:
1. In the part of simple summary, the authors state “Long-term selection or evolution is an important factor governing the differentiation of disease resistance in pigs”, but in the abstract the authors used “development” instead of “differentiation”. These two sentences should be unified, and I think “development” is better.
2. Line 51, “which is the physical basis of the immune response,” is not necessary.
3. Line 209-210, what is ” M “and “C” stand for? Please added the full name of the groups.
4. Line 376, the full name of the genes should be added under Table 1.
5. Line 416, the number of “713” should be “723”.
6. Please check all the abbreviations present in the main text to make sure they were in the form of full name for the first time.
Author Response
- In the part of simple summary, the authors said “Long-term selection or evolution is an important factor governing the differentiation of disease resistance in pigs”, but in the abstract the authors used “development” instead of “differentiation”. These two sentences should be unified, and I think “development” is better.
Response: Thanks for your suggestion. We have changed “differentiation” to “development” in “simple summary”. (Line 16).
- Line 51, “which is the physical basis of the immune response,” is not necessary.
Response:Follow your suggestion, we have deleted this sentence. (Lines 51-52).
- Line 209-210, what is ” M “and “C” stand for? Please added the full name of the groups.
Response:We have added the full name of the different pig breeds in the “Introduction” and “Materials and Methods” section. (Line 63 and Line 99).
- Line 376, the full name of the genes should be added under Table 1.
Response:Follow your suggestion, we have added the full name of the genes under Table 1. (Lines 399-400).
- Line 416, the number of “713” should be “723”.
Response:Done. (Line 446)
- Please check all the abbreviations present in the main text to make sure they were in the form of full name for the first time they occurrences.
Response:Follow your suggestion, we have carefully checked throughout the manuscript for details, and corresponding modifications have been made. (Line 110, Line 154, Line 161, Line 166, Line0169, Line 201)
Reviewer 3 Report
In this study, the authors have performed transcriptomics and proteomics analysis to detect the immune differences between Chinese and foreign pig breeds and identified critical genes and proteins involved in immune regulation. Although researchers have reached important findings here, there are few minor comments to be addressed by the authors.
1) Line 47-51, please cite the relevant literature.
2) Line 66-75, this paragraph does not make it very clear why the authors carried out this study.
3) For the introduction, authors would do well to show the link between immunity and disease resistance.
4) For the Materials and Methods, authors should indicate the manufacturer and item number of the kit, such as BCA protein kit.
5) In the result and legend of Figure 1, there is no indication of A, B, etc.
6) Lines 463-468, this sentence should be rephrased to provide more clear expression.
7) Line 482-484 and Line 485-487, please cite the relevant literature. Please check the full text carefully and correct it.
Author Response
- Line 47-51, please cite the relevant literature.
Response:We gratefully appreciate for your valuable suggestion, we have cite the relevant literature. (Line 51)
- Line 66-75, this paragraph does not make it very clear why the authors carried out this study.
Response:We gratefully appreciate for your valuable comments. Follow your and reviewer1’s suggestion, we have added more details as follow:We also have observed an interesting phenomenon that the Min pigs have lower mortality than LW pigs in our farm. And when the disease struck the farm, the growth or reproductive performance of the large white pigs were severely affected, while the Min pigs in the same enclosure remained normal. (Lines 75-78).
- For the introduction, authors would do well to show the link between immunity and disease resistance.
Response:We agree with the comment and added two references to show the link between immunity and disease resistance. (Lines 53-57)
- For the Materials and Methods, authors should indicate the manufacturer and item number of the kit, such as BCA protein kit.
Response:Done. (Line 110)
- In the result and legend of Figure 1, there is no indication of A, B, etc.
Response:Done. (Figure 1)
- Lines 463-468, this sentence should be rephrased to provide more clear expression.
Response:We agree with the comment and re-wrote the sentence in the revised manuscript as following: “As part of a self-regulatory mechanism to prevent hyperimmunity, plasma LBP is sig-nificantly increased during the acute phase of inflammation, and high concentrations of LBP could inhibit the release of LPS-mediated inflammatory cytokines, thereby inhib-iting the activation of cell signaling.” (Lines 492-495)
- Line 482-484 and Line 485-487, please cite the relevant literature. Please check the full text carefully and correct it.
Response:Done. (Lines 510-511).
Round 2
Reviewer 1 Report
Thanks for the authors' good responses! I don't have any further questions.